# Radiographic evaluation of patellar tendon length following corrective surgical procedures for medial patellar luxation in dogs

Kevin de Moya[ID]1, Stanley Kim[ID]2*

1 University of Florida College of Veterinary Medicine, Gainesville, Florida, United States of America,
2 Department of Small Animal Clinical Sciences, University of Florida College of Veterinary Medicine, Gainesville, Florida, United States of America

* stankim@ufl.edu

**Data Availability Statement:** All relevant data are within the manuscript and its Supporting Information files.

## Abstract

### Objective

To quantify changes in the patellar tendon length following surgical correction of medial patellar luxation in dogs and evaluate potential risk factors associated with patellar tendon elongation.

### Study design

Retrospective case series (n = 50).

### Methods

Dogs that underwent surgery for medial patellar luxation correction and had 2–3 months follow up were included. Digital radiographs were utilized to quantify the patellar tendon length to patellar length ratio at various follow-up points. Odds ratio comparisons between potential risk factors associated with changes in patellar tendon length were performed.

### Results

Post-operative patellar tendon lengthening of $\geq$ 5% was observed in 20% of stifles and post-operative patellar tendon shortening of $\geq$ 5% was observed in 22% of stifles at the 2–3 month follow up period. The risk factors including age, body weight, trochleoplasty and grade of medial patellar luxation were not significantly associated with risk of patellar tendon elongation. Patellar tendon lengthening was not associated with recurrence of luxation.

### Conclusion

Patellar tendon lengthening and shortening can be observed in dogs following common medial patellar luxation corrective procedures in the short term follow up period. Patellar

**Funding:** There has been no significant financial support for this work that could have influenced its outcome.

**Competing interests:** The authors have declared that no competing interests exist.

tendon lengthening does not appear to be associated with age, weight, trochleoplasty, grade of luxation, or risk of luxation recurrence.

## Introduction

Medial patellar luxation (MPL) is one of the most commonly diagnosed orthopedic diseases affecting the canine stifle [1]. Skeletal abnormalities including quadriceps displacement, distal femoral varus, genu varum, proximal tibial varus, shallow trochlear groove, coxa vara, coxa valga, hypoplastic trochlear ridges, patella alta and tibial torsion have all been associated with MPL [2–3]. Additionally, MPL has been associated with patella alta and a relatively longer patellar tendon length in medium to giant breed dogs [4]. Surgical intervention aims to properly align the quadriceps mechanism, restore normal patellar tracking within the trochlear groove, and address significant underlying skeletal abnormalities [5]. Failure to identify underlying contributing factors may lead to sub-optimal outcomes, including luxation recurrence and progression of osteoarthritis [6, 7].

Post-operative complications are associated with higher grades of luxation and include recurrence of luxation, implant migration, patellar tendinopathy, tibial tuberosity avulsion, patellar ligament rupture and infection, with post-operative luxation being the most common complication [8–11]. Given the association between patellar tendon length and MPL, it is possible that elongation of the patellar tendon following surgery could also predispose to post-operative luxation. The lengthened patellar tendon could potentially result in the patella alta, and if concurrent skeletal and soft tissue abnormalities exist, affected animals might be at an increased risk of luxation. To the author's knowledge, changes in patellar tendon length following surgical correction of MPL have not been quantified.

The purpose of this study is to radiographically quantify changes in patellar tendon length following common MPL corrective surgeries and evaluate potential risk factors that may contribute to post-operative morbidity. We hypothesize that the patellar tendon would lengthen after surgery, and would be associated with high grades of MPL (grade 3 or 4), older age, and body weight. We also hypothesized that patellar tendon lengthening might be associated with recurrence of luxation in dogs with relatively normal position of the patella.

## Material and methods

### Case selection and medical record review

Medical records were searched for dogs that had previously undergone surgical correction for medial patellar luxation between March 2008 and December 2017. All surgeries were performed by a board-certified veterinary surgeon. Dogs that had concurrent stifle disease, such as cranial cruciate ligament rupture, were excluded from this study. Dogs with lateral patella luxation were not included in the study. Dogs were included if they had appropriate radiographic follow up including pre-operative, immediately post-operative, and follow up radiographs of 2–3 months. Data collection included the animal's age, weight, grade and side of MPL, procedure(s) performed and any post-operative recurrence of luxation. Surgical treatments included tibial tuberosity transposition, distal femoral osteotomy, trochleoplasty (trochlear wedge or block recession), medial release, and lateral imbrication.

## Radiographic assessment

The evaluation of digital stifle radiographs in this study focused on the length of the patellar tendon. All measurements were performed on a standard mediolateral radiographic projection of the stifle using a dedicated PACS workstation using DICOM viewing software (Merge Healthcare Inc, Chicago, Illinois). All dogs were under general anesthesia for initial radiographs and all subsequent radiographs were performed under sedation. The radiographs were judged as satisfactory if superimposition of the femoral condyles was achieved, and if it included the metaphyseal-diaphyseal junction of both the femur and tibia. The patella length was measured as the distance from the most proximal aspect to the most distal aspect of the patella. The patellar tendon length was measured as the distance from the most distal aspect of the patella to the most proximal aspect of the tibial tuberosity (Fig 1). These measurements are consistent with those performed in previous studies that investigated MPL and associated changes in patellar tendon length in medium to giant breed dogs [4]. Care was taken to not include osteophytes when noted at the distal aspect of the patella. The patellar tendon length to patellar length ratio (PTL:PL) was then calculated. For each case, the initial digital measurements made pre-operatively were juxtaposed next to additional projections at other time points to ensure use of precisely the same landmarks for all measurements. This methodology aided in consistent measurements, particularly in cases where implants interfered with bony landmarks. Because the measurement was a ratio, calibration of the images was not necessary and thus was not performed. The angle between the femur and tibia was not measured, as previous studies have demonstrated no association between the degree of stifle flexion and two methods of calculating PTL:PL [12]. All measurements were made by one of the investigators (KD). The investigator was trained to take appropriate radiographic measurements prior to the start of the study.

## Statistical analyses

Descriptive statistics were reported as mean ± standard deviation. Unconditional odds ratios were calculated to evaluate for relationships among various potential risk factors. The specific

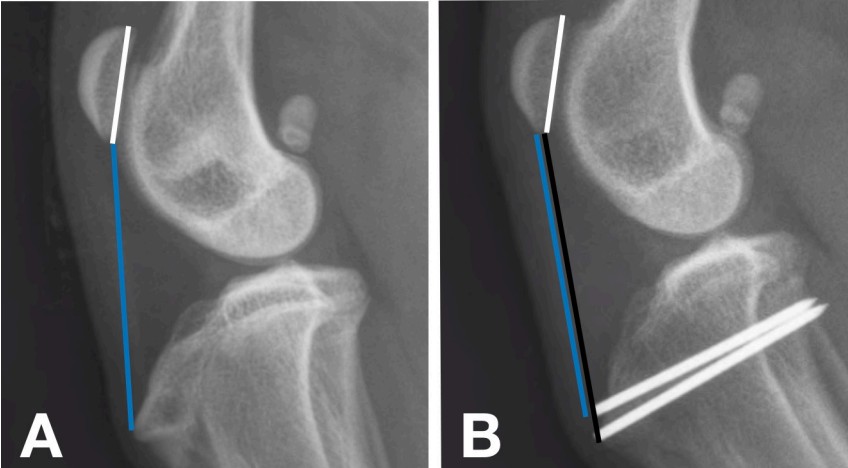

**Fig 1. Pre-operative (A) and 3 month post-operative (B) stifle radiographs in a dog with MPL correction.** Mild elongation of the patellar tendon is evident when comparing the pre-operative patellar tendon length (blue line) to the post-operative patellar tendon length (black line). Patellar tendon length: patellar length ratio was calculated by dividing the patellar tendon length over the length of the patella (white line).

odds comparisons performed included: trochleoplasty vs elongation, age (<1 year) vs elongation, age (≥1 year) vs reluxation, weight (<10 kg) vs elongation, weight (≥ 10 kg) vs reluxation, grade of MPL vs elongation, and reluxation vs elongation. For these models, elongation was defined as ≥ 5% increase in the 2–3 month postoperative PTL:PL ratio compared to preoperative values. The percentage of elongation was chosen based on previous studies that quantified PTL:PL ratios in control dogs compared to those with MPL. The study demonstrated a difference of approximately 5% in PTL:PL ratios between the upper limit of confidence interval of control dogs compared to the lower limit of the confidence interval for dogs affected by MPL [4]. In addition, this magnitude of change was well above the previously reported errors (approximately 2%) associated with the radiographic measurements. All statistics were calculated using GraphPad Prism version 8.0 (San Diego, California).

## Results

Of a total of 242 stifles surgically repaired, only 50 cases met the inclusion criteria. Mean age at the time of surgery was 2.5 ± 1.8 years, ranging from 0.5–7.7 years. Of the 50 cases, 10 dogs were less than 1 year of age and the remainder were 1 year of age or older. Mean body weight was 13.9 ± 11.0 kg ranging from 1.0 kg– 34.4 kg. Of the 50 cases, 26 dogs were less than 10 kg and the remainder weighed 10 kg or more. The distribution of MPL grade was 32% grade II, 60% grade III, and 8% grade IV. The average pre-operative PTL:PL ratio was 2.06 ± 0.32, and the average 2–3 month post-operative PTL:PL ratio was 2.05 ± 0.29. Of the 50 cases, 46% of dogs had patellar tendon lengthening and 54% of dogs had shortening at the 2–3 month follow up point. Of the 50 cases, 10 of the dogs had patellar tendon lengthening of ≥ 5% and 11 of the dogs had patellar tendon shortening of ≥ 5% at the 2–3 month follow up point. Age, body weight, and grade of MPL were not significantly associated with risk of patellar tendon elongation (Table 1). Patellar tendon lengthening was not associated with recurrence of MPL (Table 1). The distribution of percent change in PTL:PL ratios from pre-operative to final follow up measurements are included in Fig 2.

## Discussion

The patellar tendon is subjected to changes following surgical procedures involving the stifle. Retrospective studies in human have demonstrated an increased prevalence of patellar tendon shortening with total knee arthroplasties and increased prevalence of patellar tendon lengthening with unicompartmental knee arthroplasty at 5 years post-surgery [13]. Previous investigators have demonstrated reduced blood flow to the patellar tendon following TPLO and patellar luxation procedures in cadaveric canine models [14]. Previous studies have established that large breed dogs with MPL have a relatively longer patellar tendon compared to dogs without stifle disease [4]. Other investigators have demonstrated shortening of patellar tendon

**Table 1. Odds ratio calculations for potential risk factors associated with patellar tendon elongation and recurrence of patellar luxation.**

| Parameter | Odds Ratio | 95% Confidence interval | Z statistic | P-value |
|---|---|---|---|---|
| Trochleoplasty vs elongation | 0.3382 | 0.0636–1.7995 | 1.271 | 0.2037 |
| Age (<1 year) vs elongation | 0.5 | 0.1169–2.1392 | 0.935 | 0.35 |
| Age (≥1 year) vs reluxation | 0.4355 | 0.1005–1.8879 | 1.111 | 0.2666 |
| Weight (<10 kg) vs elongation | 0.3878 | 0.0876–1.7172 | 1.248 | 0.2121 |
| Weight (≥10 kg) vs reluxation | 0.2045 | 0.0385–1.0866 | 1.862 | 0.0625 |
| Grade of MPL vs elongation | 0.6429 | 0.1531–2.6986 | 0.604 | 0.5461 |
| Reluxation vs elongation | 2.4287 | 0.4869–12.1138 | 1.082 | 0.2792 |

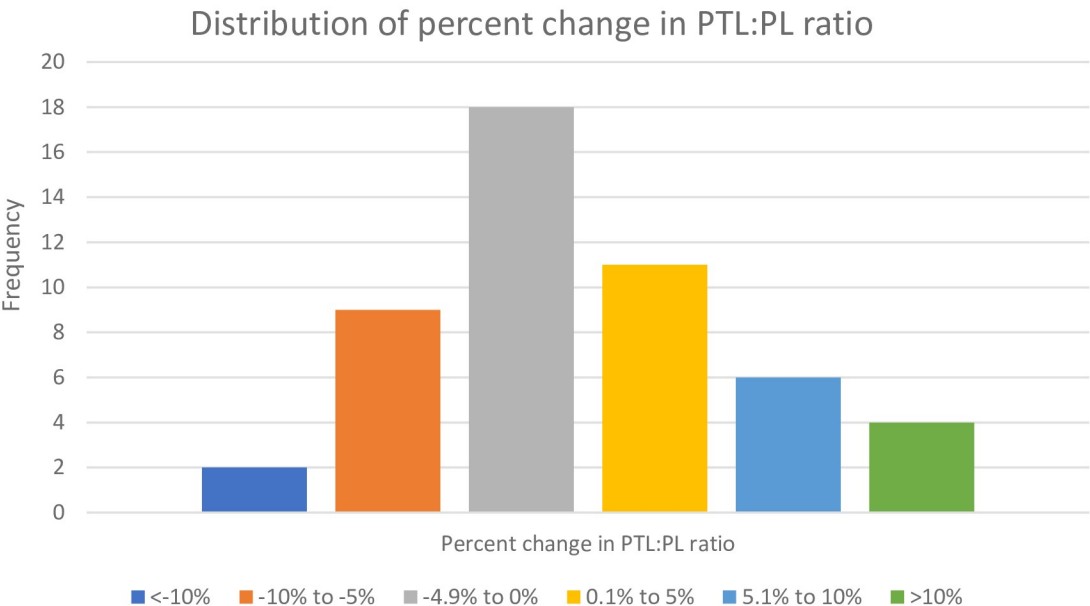

**Fig 2. Distribution of percent change in PTL:PL ratios.** The percent change was calculated using pre-operative PTL:PL measurements compared to final follow-up measurements.

length following tibial plateau leveling osteotomy procedures in dogs [15]. Although patella alta has previously been proposed as predisposing factor to post-operative recurrence of luxation, other studies have demonstrated this condition in healthy dogs without evidence of orthopedic disease [12, 16]. These factors in addition to other underlying soft tissue and skeletal abnormalities may all contribute to changes in patellar tendon length.

The main findings of our study show that patellar tendon length can increase or decrease following MPL procedures. However, these observations were not significantly associated with the any of the risk factors we investigated, such as age, body weight and grade of MPL. Furthermore, there was no association between postoperative lengthening of the patellar tendon with luxation recurrence.

For the purposes of our investigation, we defined elongation as a 5% or greater increase in the PTL:PL ratio. The 95% CI of the PLL:PL ratio observed in previous studies was 1.92–2.03 for large breed dogs with MPL [4]. This is similar to our average pre-operative PLL:PL ratio of 2.06, and our average 2–3 month follow up ratio of 2.05. Since our study population included both large and small breed dogs, it may be more representative of the broader population of dogs affected by MPL when compared to previous investigations [4].

Dogs are considered skeletally mature by 1 year of age and at this point the structures of the quadriceps mechanism should be completely developed. Therefore, we investigated the association between <1 year of age and risk of elongation >5%, and the association between >1 year of age and risk of luxation recurrence. Although reluxation was noted in some cases, there was not a significant association between age > 1 year and risk of patellar reluxation. These findings are consistent with other retrospective studies that found no association between age and risk for developing complications following MPL procedures [8]. Although the PTL:PL ratio increased in 23/50 cases at 2–3 months post-operatively, patellar elongation did not appear to be influenced by age in our study.

Recurrence of patellar luxation and other post-operative complications following MPL corrective surgeries have been associated with in higher grade MPL, where there may be more

tension on the quadriceps mechanism after surgery [8–10]. Of the 50 included cases, 9 had post-operative recurrence of luxation. Of these 9 cases with post-operative patellar luxation, three dogs had an MPL grade of 2, three dogs had an MPL grade of 3, and one dog had an MPL grade of 4. For the purposes of our investigation, we defined higher grades of MPL as grade 3 or 4. Although there was lengthening observed in higher grades of MPL, there was no significant association between grade of MPL and risk of elongation >5%. This may be due to the fact that the amount of lengthening that occurred was not substantial enough to have the patellar positioned proximal to the trochlear groove which contributes to the potential for recurrence of luxation. Another consideration could be pre-existing elongation of the patellar tendon that is subjected to abnormal tension during development and physical activity. Based on these findings, a long patellar tendon appears to be more clinically relevant if it is pre-existing, rather than as a consequence of surgery.

Other investigators have also detected a correlation between body weight and risk of recurrence of luxation following MPL corrective procedures [9, 11]. The average body weight of dogs included in our investigation was lower than those reported in other studies investigating PLL:PL [4, 12]. This is due to the fact that the aforementioned studies only included populations of large breed dogs, while our study population included both large and small breed dogs. Although there was observed lengthening as well as shortening, there was no significant association between weight small (< 10 kg) vs medium-large dogs (>10 kg) and risk of elongation. Additionally, there was no association between size and risk of recurrence of luxation.

The retrospective nature of this investigation was a limitation of this study. This could have contributed to some inaccuracies regarding the presence of lower grade luxation recurrence or other mild post-operative complications. The short period of follow-up of 2–3 months is another limitation. The lack of radiographic follow-up was the main reason many cases did not qualify for this study. Future prospective studies could include longer term follow up to assess the change in PLL:PL ratios over time in conjunction with more accurate clinical assessment. Another limitation of the study was the inability of blinding of the images when acquiring the measurements. However, images were compared within each specimen to ensure consistent landmarks were used. Measurements were only performed once per image by a single observer, and use of multiple measurements may have improved the accuracy of the study. This was not deemed necessary because a previous study demonstrated low intra- and inter-observer variability of PLL [4, 15]. Other limitations include the lack of control groups for comparison and the wide breed variability. In this study, dogs were categorized based on weight and specific breeds were not taken into account. Future studies could assess whether patellar elongation is more clinically relevant in specific breeds.

Lengthening and shortening of the patellar tendon can occur both small and large breed dogs following common MPL procedures. Our investigation revealed that these changes are not associated with age, body weight or grade of MPL, nor do they influence the risk of luxation recurrence in the short-term. Further investigation of these parameters is warranted to characterize the changes in patellar tendon length over a longer follow-up period.

## Supporting information

**S1 Table.**
(XLSX)

## Author Contributions

**Conceptualization:** Kevin de Moya, Stanley Kim.

**Data curation:** Kevin de Moya.

**Formal analysis:** Kevin de Moya.

**Investigation:** Kevin de Moya.

**Resources:** Stanley Kim.

**Software:** Stanley Kim.

**Supervision:** Stanley Kim.

**Writing – original draft:** Kevin de Moya.

**Writing – review & editing:** Kevin de Moya.

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
