## [Decision Letter · Decision Letter 0]

11 Jun 2020

PONE-D-20-13783

Radiographic evaluation of patellar tendon length following corrective surgical procedures for medial patellar luxation in dogs.

PLOS ONE

Dear Dr. de Moya,

Thank you for submitting your manuscript to PLOS ONE. After careful consideration, we feel that it has merit but does not fully meet PLOS ONE’s publication criteria as it currently stands. Therefore, we invite you to submit a revised version of the manuscript that addresses the points raised during the review process.

We look forward to receiving your revised manuscript.

Kind regards,

Silvia Sabattini

Academic Editor

PLOS ONE

2. Thank you for including your funding statement; "The funders had no role in study design, data collection and analysis, decision to publish, or preparation of the manuscript."

3. Please amend the manuscript submission data (via Edit Submission) to include author Stanley Kim

Reviewers' comments:

Reviewer's Responses to Questions

**Comments to the Author**

1. Is the manuscript technically sound, and do the data support the conclusions?

Reviewer #1: Yes

Reviewer #2: Yes

2. Has the statistical analysis been performed appropriately and rigorously? 

Reviewer #1: Yes

Reviewer #2: N/A

3. Have the authors made all data underlying the findings in their manuscript fully available?

Reviewer #1: Yes

Reviewer #2: Yes

4. Is the manuscript presented in an intelligible fashion and written in standard English?

Reviewer #1: Yes

Reviewer #2: Yes

5. Review Comments to the Author

Reviewer #1: Dear Authors, I have been really pleased to review this manuscript. I think patellar luxation is one of the most important disease that any orthopaedic surgeon has to deal with. Although we have a huge kwnoledge of this condition, I belive there is still a lot of to understand to improve our diagnosis, treatment and outcome. For this reason any information that might help reaching this goal is very valuable.

Regarding the study presented here, looking at the changings of the patellar tendon and its effect on the vertical position of the patella, I think this is an interesting point of view that has still not been properly studied.

I don't have many critics to do about the aim of the paper, the statistical analysis and the contents in general. I have given some suggestions to try to improve the text, and I have tried to find potential conflicts that need to be addressed to make the manuscript sound. My biggest critic is regarding references since PL is a quite popular topic in veterinary literature and when I read a paper on this topic I expect to find a lot of scientific support.

I would also consider to find some evidence of the same condition in human literature, it might be interesting for comparing your results.

My general judgment is positive. Following you can find some specific comments. I hope my suggestions will help you in publishing this paper.

COMMENTS:

39 – correct “patellar” to “patella”

40-42 – Mostafa 2008 seems to be the only paper used as reference for this study. See comments later.

47-50 – more references.

50-52 – confusing, reporting some info as on line 40-42

52-54 – this is probably non-relevant information. The paper deals with MPL and any other affection of the stifle lead to exclusion. I would erase it.

54-56 – please discuss, for a better understanding of the Reader, why this can happen.

62-63 – please change “would” to “might”. Your hypothesis is correct from my point of view; however have you considered that some dogs, especially some small breed, can have concomitant MPL and patella baja, and in this case the elongated patellar tendon should improve the tracking and not increase the risk of luxation. Maybe you should specify that the hypothesis is limited to dog with a relatively normal position of the patella.

71 – is 2-3 months an appropriate follow up? Patellar tendon stretching and elongation might take longer, also some complications as reluxation might happen later in time.

71-73 – please remove “from medical records”; correct “include” to “included”; please remove “comorbidities” because you stated previously your inclusion/exclusion criteria; change “surgical procedure(s) and post-operative complications” (let’s follow an appropriate chronologic order)

73 – Please change “Surgical corrective procedures performed include” to “Surgical treatment included”

78 – the reference to the Figure should be moved at the end of the sentence that describes the methodology of measurement.

77-80 – I am not convinced of the first sentence, it does not sound well. I would suggest to change this part as follow “All measurements were performed on a standard mediolateral radiographic projection of the stifle by using a dedicated PACS workstation using DICOM viewing software (Merge Healthcare Inc, Chicago, Illinois).”

80-81 – Can you mention the angle between femur and tibia? Do you think that limb positioning (in this case the degree of flexion) may affect measurement and result? (for replicating your study it is important to give to the Reader all the relevant information).

Maybe this can be helpful: “There was no association between the degree of stifle joint flexion and PLLj/PL and PLLm/PL ratios, respectively” (Łojszczyk-Szczepaniak, A., Silmanowicz, P., Komsta, R. et al. Determination of reference values and frequency of occurrence of patella alta in German shepherd dogs: a retrospective study. Acta Vet Scand 59, 36 (2017). https://doi.org/10.1186/s13028-017-0304-1)

82 – remove “from”, written twice

82-86 – vertical position of the patella in dogs has been previously reported and several methods have been described (for example Mostafa et al. and Johnson et al. described two different measurements). Can you please add a reference to your methodology in order to help the Reader to understand that not all the published studies are comparable between each other?

86-88 – this is a good way to reduce the risk of error. My recommendation for next time for more reliable data is to make repeated measurement and check for intraobserver difference in order to be sure that final number is as much precise as possible.

88 – regarding landmarks, have you noticed any implant interference in identifying the correct landmark? The accuracy of measurement might be discussed.

110 – “On a total of 242 stifles surgically repaired, only 50 cases met the inclusion criteria”. Maybe sounds better.

111 – “The majority of cases were excluded due to inadequate radiographic follow up”. I think you can delete this sentence since you stated previously your inclusion/exclusion criteria.

134 – Here I would mention a sentence that I suggested to remove in the introduction regarding the reported changings following TPLO procedure. I would start the discussion stating that the patellar tendon is subjected to changings and that some pathologies as well treatments can affect its length. Try to find further references (also check human literature) to strengthen this statement because this is the pillar of the paper.

139-142 – Please check the literature, there are papers that provide some information that are in contrast with studies you cited. It is important in the discussion review all the available literature to find evidence. “Additionally, patella alta has previously been proposed as predisposing factor to post-operative recurrence of luxation (Johnson 2006)”, but patella alta has also been observed in healthy dogs that did not exhibit orthopaedic problems in the stifle joints (Łojszczyk-Szczepaniak, A., Silmanowicz, P., Komsta, R. et al. Determination of reference values and frequency of occurrence of patella alta in German shepherd dogs: a retrospective study. Acta Vet Scand 59, 36 (2017)).

If you do not clarify this statements it sounds like that if the patella is alta you can have reluxation. I believe this complication is more related to other abnormalities rather than a patella higher than normal.

Wandagee 2013 reported “The outcome of surgery was considered good for grade II luxation with a 100% success rate. Recurrent medial patellar luxation was diagnosed in approximately 11% of dogs with grade III and in 36% of dogs with grade IV luxation.”

By reading the available literature, we have evidence that risk of reluxation is relatively low and patella alta might not be as relevant as you stated. And reluxation is generally observed in dogs with severe stifle abnormalities (both skeletal and of the soft tissues)

“Previous studies have established that large breed dogs with MPL have a relatively long patellar tendon compared to dogs without stifle disease (Mostafa 2008)”. This statement is correct, but remember that MPL is typically seen in small breed dogs.

160 – Please add more references.

161-167 – This is probably one of the most interesting part of the paper. You must answer this question “Why grade 3-4 MPL have not significantly different ratios?”. This is probably due to the pre-existing elongation of the tendon that is subjected to abnormal tension during growth or normal physical activity.

177-190 – Very good paragraph that answer some of my previous comments. Last sentence in incomplete.

Reviewer #2: Dear authors this topic is very interesting, and the study is well done but some issues must be revised. please find attached the comments.

Comment 1) Please add the name of the second co-author (all co-authors) at the beginning, in the "Order of Authors".

Comment 2) According to the journal guidelines, the references should be cited as the numbers in brackets. They should be numbered in the order they appear in the text. Please revise your citations.

Comment 3) According to the journal guidelines, the abstracts should not include abbreviations if it is possible.

Comment 4) Line 39: Please revise the words "patellar alta" to "patella alta".

Comment 5) Line 68-69: You did not talk about LPL in your study. Please explain if you excluded the dogs with LPL from your study. If LPL affected dogs were not excluded please explain the dogs with MPL and LPL. Is there any relationship between the dogs with bidirectional patellar luxation and post-operative patellar tendon length?

Comment 6) Please talk about the muscle contracture in your study and if the radiographs were taken under general anesthesia.

Comment 7) Lines 84 – 85. Please revise the sentence "the distance from the distal most aspect of the patellar to the proximal aspect of the tibial tuberosity" as "the distance from the most distal aspect of the patella to the most proximal aspect of the tibial tuberosity".

Comment 8) Lines 93 - 94. Please revise the sentence "Patella tendon: patellar ratio" as "Patellar tendon length: patellar length ratio".

Comment 9) Please include the name and version of the used statistical software and calculation methods (name of the tests) in the material and methods.

Comment 10) Please discuss the included dog breeds in your study. You only mentioned in the discussion that you had small to large breed dogs; it would not be out of interest if you could explain the frequency of each breed.

Comment 11) Line 158. You are talking about post-operative complications reported in the literature. Did you have any post-operative complications in included dogs? Please discuss it.

Comment 12) Line 185. Please discuss the level of experience of the observer who measured the alignments. Did the observer train to measure alignments before the study?

Comment 13) Line 190. The sentence "another limitation of the study was" is incomplete, please delete it.

Best regards

6. PLOS authors have the option to publish the peer review history of their article (what does this mean?). If published, this will include your full peer review and any attached files.

Reviewer #1: No

Reviewer #2: No

---

## [Author Response · Author response to Decision Letter 0]

9 Jul 2020

Dear PLOS ONE Editors and Reviewers, 

Thank you for the critiques and suggestions for our manuscript. The majority of the recommendations made have added clarification and quality to the discussion of our study. We feel that the revised manuscripts have adequately addressed the concerns raised during the review process. Please see review the following rebuttal letter that addresses suggestions made. We hope that the revised manuscript is suitable for publication and look forward to hearing from you in due course. 

Best, Kevin de Moya 

1. Please ensure manuscript meets PLOS ONE’s style requirements including those for file naming

• The manuscript formatting has been updated including references and requirements made for file naming. 

2. Please include your amended funding statements within your cover letter.

• See updated cover letter. 

3. Please amend the manuscript submission data (via Edit Submission) to include author Stanley Kim.

• See amended manuscript.

4. Please include captions for your Supporting Information files at the end of your manuscript, and update any in-text citations to match accordingly.

• See amended manuscript.

5. Review comments to Author (Reviewer #1)

• Comment- Correct “patellar” to “patella” (line 39).

• Response- As suggested, this correction has been made (line 39).

• Comment- Add more references (line 47-50).

• Response- As suggested, additional references have been added to support the statements made (line 46-50). 

• Comment- Confusing, reporting some info as on line 40-42 (line 50-52).

• Response- This sentence has been deleted. 

• Comment- This is probably non-relevant information. The paper deals with MPL and any other affection of the stifle lead to exclusion. I would erase it (line 52-54). 

• Response- As suggested, this sentence has been deleted. 

• Comment- Please discuss, for a better understanding of the Reader, why this can happen (line 54-56).

• Response- As suggested, an additional sentence has been added to further clarify the statements made (line 51-53).

• Comment- Please change “would” to “might”. Your hypothesis is correct from my point of view; however have you considered that some dogs, especially some small breed, can have concomitant MPL and patella baja, and in this case the elongated patellar tendon should improve the tracking and not increase the risk of luxation. Maybe you should specify that the hypothesis is limited to dog with a relatively normal position of the patella (line 62-63).

• Response- As suggested, an addition to the hypothesis was added clarifying that our hypotheses apply to dogs with relatively normal patella positions (57-60).

• Comment- Is 2-3 months an appropriate follow up? Patellar tendon stretching and elongation might take longer, also some complications as reluxation might happen later in time (line 71). 

• Response- 2-3 months is the last radiographic follow up time point before animals are discharged to resume normal activity, assuming no complications occurred. Although we suspect that patellar tendon lengthening can occur past this time point, we do not enough cases with radiographic follow up past this time point. 

• Comment- Please remove “from medical records”; correct “include” to “included”; please remove “comorbidities” because you stated previously your inclusion/exclusion criteria; change “surgical procedure(s) and post-operative complications” (line 71-73).

• Response- The statement has been updated to reflect suggestions made (line 69-71).

• Comment- Please change “Surgical corrective procedures performed include” to “Surgical treatment included” (line 73).

• Response- The statement has been updated to reflect suggestions made (line 70-71).

• Comment- The reference to the Figure should be moved at the end of the sentence that describes the methodology of measurement (line 78).

• Response- The reference to Figure 1 has been moved to the end of the sentence that describes the measurement methodology (line 83).

• Comment- I am not convinced of the first sentence, it does not sound well. I would suggest to change this part as follow “All measurements were performed on a standard mediolateral radiographic projection of the stifle by using a dedicated PACS workstation using DICOM viewing software (Merge Healthcare Inc, Chicago, Illinois)- (line 77-80).

• Response- The statement has been updated to reflect suggestions made (line 75-77).

• Comment- Can you mention the angle between femur and tibia? Do you think that limb positioning (in this case the degree of flexion) may affect measurement and result? (for replicating your study it is important to give to the Reader all the relevant information)- (line 80-81).

• Response- An additional statement has been added with proper reference justifying why the degree of stifle flexion was not included in the study (92-94).

• Comment- Vertical position of the patella in dogs has been previously reported and several methods have been described (for example Mostafa et al. and Johnson et al. described two different measurements). Can you please add a reference to your methodology in order to help the Reader to understand that not all the published studies are comparable between each other? (line 82-86).

• Response- The statement has been updated to reflect that the measurements made are consistent with those of previous studies (83-85).

• Comment- Regarding landmarks, have you noticed any implant interference in identifying the correct landmark? The accuracy of measurement might be discussed (line 88).

• Response- An additional statement has been added describing the methodology used in cases where there was implant interference with bony landmarks (89-91). 

• Comment- “On a total of 242 stifles surgically repaired, only 50 cases met the inclusion criteria”. Maybe sounds better (line 110).

• Response- The statement has been updated to reflect suggestions made (line 117). 

• Comment- “The majority of cases were excluded due to inadequate radiographic follow up”. I think you can delete this sentence since you stated previously your inclusion/exclusion criteria (line 111).

• Response- As suggested, this statement has been deleted. 

• Comment- Here I would mention a sentence that I suggested to remove in the introduction regarding the reported changings following TPLO procedure. I would start the discussion stating that the patellar tendon is subjected to changings and that some pathologies as well treatments can affect its length. Try to find further references (also check human literature) to strengthen this statement because this is the pillar of the paper (line 134). 

• Response: This paragraph has been updated to reflect both veterinary and human studies that investigate changes in patellar tendon length as an introduction to our discussion. The veterinary studies that were previous mentioned in the introduction were moved to this section (141-153).

• Comment- Please check the literature, there are papers that provide some information that are in contrast with studies you cited. It is important in the discussion review all the available literature to find evidence. “Additionally, patella alta has previously been proposed as predisposing factor to post-operative recurrence of luxation (Johnson 2006)”, but patella alta has also been observed in healthy dogs that did not exhibit orthopaedic problems in the stifle joints (Łojszczyk-Szczepaniak, A., Silmanowicz, P., Komsta, R. et al. Determination of reference values and frequency of occurrence of patella alta in German shepherd dogs: a retrospective study. Acta Vet Scand 59, 36 (2017)). If you do not clarify this statements it sounds like that if the patella is alta you can have reluxation. I believe this complication is more related to other abnormalities rather than a patella higher than normal (line 139-142).

• Response- As suggested, the statement has been updated with further clarification and additional references (line 149-153).

• Comment- Add more references (line 160).

• Response- As suggested, this paragraph has been updated with additional references (line 176). 

• Comment- This is probably one of the most interesting part of the paper. You must answer this question “Why grade 3-4 MPL have not significantly different ratios?”. This is probably due to the pre-existing elongation of the tendon that is subjected to abnormal tension during growth or normal physical activity (Line 161-167).

• Response- As suggested, this paragraph has been updated to give a proposal of why grade 3-4 MPL may not have significantly different ratios (181-183).

6. Review comments to Author (Reviewer #2)

• Comment- Please add the name of the second co-author (all co-authors) at the beginning, in the “order of authors”.

• Response- The name of the second co-author has been added in the “order of authors” section. 

• Comment- According to the journal guidelines, the references should be cited as the numbers in brackets. They should be numbered in the order they appear in the text. Please revise your citations.

• Response- The references have been updated and are cited as numbers within brackets and are in the order they appear in the text. 

• Comment- According to the journal guidelines, the abstracts should not include abbreviations if it is possible.

• Response- The abstract has been updated and does not include any abbreviations. 

• Comment- Please revise the word “Patellar alta” to “Patella alta” (line 39).

• Response- As suggested, the statement has been updated to reflect suggestions made (line 39).

• Comment- You did not talk about LPL in your study. Please explain if you excluded the dogs with LPL from your study. If LPL affected dogs were not excluded please explain the dogs with MPL and LPL. Is there any relationship between the dogs with bidirectional patellar luxation and post-operative patellar tendon length? (line 68-69).

• Response- A statement has been added to the Methods section that clarifies dogs with lateral patellar luxation were excluded from the study (line 66-67).

• Comment- Please talk about the muscle contracture in your study and if the radiographs were taken under general anesthesia.

• Response- As suggested, an additional statement has been added regarding anesthesia and sedation for radiographs (line 77-78). 

• Comment- Please revise the sentence "the distance from the distal most aspect of the patellar to the proximal aspect of the tibial tuberosity" as "the distance from the most distal aspect of the patella to the most proximal aspect of the tibial tuberosity" (line 84-85).

• Response: The statements have been updated to reflect suggestions made (line 80-83).

• Comment- Please revise the sentence "Patella tendon: patellar ratio" as "Patellar tendon length: patellar length ratio" (line 93-94).

• Response- The statement has been updated to reflect suggestions made (line 86-87). 

• Comment- Please include the name and version of the used statistical software and calculation methods (name of the tests) in the material and methods.

• Response- A statement has been added to reflect the suggestions made (line 114-115).

• Comment- Please discuss the included dog breeds in your study. You only mentioned in the discussion that you had small to large breed dogs; it would not be out of interest if you could explain the frequency of each breed.

• Response- Dogs were categorized based on weight (<10kg vs >10kg) and specific breeds were not recorded. The sentence has been updated to accurately reflect data collected on each animal (line 69-70).

• Comment- Please discuss the level of experience of the observer who measured the alignments. Did the observer train to measure alignments before the study? (line 185).

• Response- An updated statement in the methods section is included to describe the training involved in taking radiographic measurements prior to the start of the study (line 94-96).

• Comment- You are talking about post-operative complications reported in the literature. Did you have any post-operative complications in included dogs? Please discuss it (line 158). 

• Response- Additional sentences have been added that describe the number of dogs with post-operative patellar luxation and the associated grade of MPL for each dog (line 176-179).

• Comment- The sentence "another limitation of the study was" is incomplete, please delete it (line 190).

• Response- As suggested, this statement has been deleted.

---

## [Decision Letter · Decision Letter 1]

12 Aug 2020

PONE-D-20-13783R1

Radiographic evaluation of patellar tendon length following corrective surgical procedures for medial patellar luxation in dogs.

PLOS ONE

Dear Dr. Kim,

Thank you for submitting your manuscript to PLOS ONE and for this revised version. The reviewers have raised a few minor criticisms that can be easily addressed prior to publication.

We look forward to receiving your revised manuscript.

Kind regards,

Silvia Sabattini

Academic Editor

PLOS ONE

Reviewers' comments:

Reviewer's Responses to Questions

**Comments to the Author**

1. If the authors have adequately addressed your comments raised in a previous round of review and you feel that this manuscript is now acceptable for publication, you may indicate that here to bypass the “Comments to the Author” section, enter your conflict of interest statement in the “Confidential to Editor” section, and submit your "Accept" recommendation.

Reviewer #1: All comments have been addressed

Reviewer #2: (No Response)

2. Is the manuscript technically sound, and do the data support the conclusions?

Reviewer #1: Yes

Reviewer #2: Yes

3. Has the statistical analysis been performed appropriately and rigorously? 

Reviewer #1: N/A

Reviewer #2: Yes

4. Have the authors made all data underlying the findings in their manuscript fully available?

Reviewer #1: Yes

Reviewer #2: Yes

5. Is the manuscript presented in an intelligible fashion and written in standard English?

Reviewer #1: Yes

Reviewer #2: Yes

6. Review Comments to the Author

Reviewer #1: Thank you for this revised version of the paper. All the previous comments have been addressed and now the paper is sound.

Minor comments:

23 - at least 1 month follow up; line 69, 108 is 2-3 months follow up. Although you say in the abstract "at least one month" and it is possible that none of the patients had only 1 month fu, it sounds strange. You could change in the text "2-3 months fu" with "the last fu" since doesn't matter if it is 2-3 months or longer or a bit lesser because in the inclusion criteria you stated "at least one month fu".

147 – change long to longer

206-207 – please cite the study

Reviewer #2: Dear Authors,

Thank you very much for submitting the revised manuscript. I do see that the requested edits were made and submitted successfully in the Editorial Manager. However, there are still some outstanding issues that must be resolved. Please revise the following parts.

Line 83) Please revise the word ''patellar'' to the ''patella''.

Line 178) Please revise the word ''glad'' to the word ''grade''.

Kind regards

7. PLOS authors have the option to publish the peer review history of their article (what does this mean?). If published, this will include your full peer review and any attached files.

Reviewer #1: No

Reviewer #2: No

---

## [Author Response · Author response to Decision Letter 1]

12 Aug 2020

Thank you. All suggested changes have been made.

---

## [Editor Report · Decision Letter 2]

20 Aug 2020

Radiographic evaluation of patellar tendon length following corrective surgical procedures for medial patellar luxation in dogs.

PONE-D-20-13783R2

Dear Dr. Kim,

We’re pleased to inform you that your manuscript has been judged scientifically suitable for publication and will be formally accepted for publication once it meets all outstanding technical requirements.

Kind regards,

Silvia Sabattini

Academic Editor

PLOS ONE
---

## [Editor Report · Acceptance letter]

26 Aug 2020

PONE-D-20-13783R2 

Radiographic evaluation of patellar tendon length following corrective surgical procedures for medial patellar luxation in dogs. 

Dear Dr. Kim:

I'm pleased to inform you that your manuscript has been deemed suitable for publication in PLOS ONE. Congratulations! Your manuscript is now with our production department. 

Kind regards, 

on behalf of

Dr. Silvia Sabattini 

Academic Editor

PLOS ONE